# Biological Control of Charcoal Rot in Peanut Crop through Strains of *Trichoderma* spp., in Puebla, Mexico

**DOI:** 10.3390/plants10122630

**Published:** 2021-11-30

**Authors:** Saira Jazmín Martínez-Salgado, Petra Andrade-Hoyos, Conrado Parraguirre Lezama, Antonio Rivera-Tapia, Alfonso Luna-Cruz, Omar Romero-Arenas

**Affiliations:** 1Facultad de Ciencias Biológicas, Programa Biotecnología, Benemérita Universidad Autónoma de Puebla, Ciudad Universidad Autónoma de Puebla, Ciudad Universitaria, Puebla 72570, Mexico; jazmin_saira@hotmail.com; 2Centro de Agroecología, Instituto de Ciencias, Benemérita Universidad Autónoma de Puebla, Edificio VAL 1, Km 1.7 Carretera a San Baltazar Tetela, San Pedro Zacachimalpa, Puebla 72960, Mexico; andrad@colpos.mx (P.A.-H.); conrado.parraguirre@correo.buap.mx (C.P.L.); 3Centro de Investigaciones en Ciencias Microbiológicas, Instituto de Ciencias, Benemérita Universidad Autónoma de Puebla, Ciudad Universitaria, Puebla 72570, Mexico; jose.riverat@correo.buap.mx; 4Instituto de Investigaciones Químico-Biológicas, Universidad Michoacana de San Nicolas de Hidalgo, Morelia 27852, Mexico; aluna@colpos.mx

**Keywords:** charcoal rot, biological control, percentage of radial growth inhibition (PIRG), incidence of the disease, production

## Abstract

Charcoal rot is an emerging disease for peanut crops caused by the fungus *Macrophomina phaseolina*. In Mexico, peanut crop represents an important productive activity for various rural areas; however, charcoal rot affects producers economically. The objectives of this research were: (a) to identify and morphologically characterize the strain “PUE 4.0” associated with charcoal rot of peanut crops from Buenavista de Benito Juárez, belonging to the municipality of Chietla in Puebla, Mexico; (b) determine the in vitro and in vivo antagonist activity of five *Trichoderma* species on *M. phaseolina*, and (c) determine the effect of the incidence of the disease on peanut production in the field. Vegetable tissue samples were collected from peanut crops in Puebla, Mexico with the presence of symptoms of charcoal rot at the stem and root level. The “PUE 4.0” strain presented 100% identity with *M. phaseolina*, the cause of charcoal rot in peanut crops from Buenavista de Benito Juárez. *T. koningiopsis* (T-K11) showed the highest development rate, the best growth speed, and the highest percentage of radial growth inhibition (PIRG) over *M. phaseolina* (71.11%) under in vitro conditions, in addition, *T. koningiopsis* (T-K11) showed higher production (1.60 ± 0.01 t/ha^−1^) and lower incidence of charcoal rot under field conditions. The lowest production with the highest incidence of the disease occurred in plants inoculated only with *M. phaseolina* (0.67 ± 0.01 t/ha^−1^) where elongated reddish-brown lesions were observed that covered 40% of the total surface of the main root.

## 1. Introduction

Peanut (*Arachis hypogea* L.) is a self-pollinated annual tropical legume that belongs to Papilionaceae subfamily, native to South America and valued worldwide for its high content of oil, proteins, and minerals such as iron, calcium, phosphorus, magnesium, selenium, and zinc; in addition to vitamins E, B6, riboflavin, thiamine, and niacin [1,2]. China is the world’s leading producer of peanuts, accounting for nearly 41.0% of the total output. In 2019, China was the biggest peanut producer with a production of 17.5 million metric tons. India, Nigeria, and the United States followed with about 6.8, 3.0, and 2.5 million metric tons, respectively [3]. The cultivated area in Mexico is currently 47,532 ha with a production of 81,413 tons in 2019 [4]. State of Puebla ranks third in national production with 9.31 tons [5].

Peanut crop can be affected by various diseases caused by fungi, bacteria, and viruses that affect the yield, fungal diseases being the most worrying that generate significant economic losses [6]. Among fungal diseases of peanuts, charcoal rot is a disease caused by *Macrophomina phaseolina*, recently reported in Mexico [7]. However, charcoal rot affects more than 500 economically important plant species, such as cotton (*Gossypium hirsutum*), the chickpea (*Cicer Arientitanium*), the beans (*Phaseolus vulgaris*), the potato (*Solanum tuberosum*), the soybean (*Glycine max*), the corn (*Zea mays*), and the peanuts (*A. hypogaea*) [8,9].

*M. phaseolina* is a generalist phytopathogenic fungus originating in the soil and in the seed, present throughout the world [10]. It is characterized by hyaline hyphae with thin walls and light brown or dark brown with septa. Microsclerotia form a compact mass of hardened fungal mycelium that darkens with aging [11]. The *M. phaseolina* phytopathogen can infect the roots of the host plant at the seedling stage through multiple germinating hyphae. Once in the roots, the hyphae affect the vascular system, interrupting the transport of water and nutrients to the upper parts of the plants, causing the yellowing and senescence of the leaves. Charcoal rot mainly affects the lower stem and main root, causing premature death of the host plant [12].

There are several effective fungicides available and labeled for use in peanut (*A. hypogaea*) crop to control various fungal diseases, including demethylation inhibitors, growth inhibitors, and succinate dehydrogenase inhibitors (codes 3, 11, and 7 of the Fungicide Resistance Action Committee, respectively) [13]. Faced with the high costs of fungicides and their potentially harmful effects on people and environment, biological control is considered viable practice for development of sustainable agriculture [14,15].

Use of filamentous fungi as biocontrol agents represents an effective alternative for agricultural production systems [16]. Success and use in agroecological practice are due to its action mechanisms such as competition for space, mycoparasitism, antibiosis [17,18], and production of volatile compounds [19]. Management of charcoal rot by means of antagonistic microorganisms, such as *Pseudomonas fluorescens* and *Trichoderma* spp., has been carried out in economically important crops, among which are soybean [20], the sorghum [21], the beans, and sunflower [22]. However, the control of charcoal rot remains a challenge despite the many efforts that have been made about research. Therefore, the objective of this research was to evaluate antagonistic capacity of five *Trichoderma* species against “PUE 4.0” strain, present in peanut crop through in vitro and field tests in rural communities in Buenavista de Benito Juárez, belonging to the municipality of Chietla in Puebla, Mexico.

## 2. Materials and Methods

### 2.1. Area of Isolation

Vegetable tissue samples with rot at the stem and root level were collected in a plot of 3144.3 m^2^ of peanut crop with a history of high charcoal rot incidence [7] during the summer–fall 2020 production. Agricultural plot corresponds to the Buenavista de Benito Juárez community, belonging to the municipality of Chietla in the state of Puebla-Mexico, with a warm desert climate (Bwh) and average rainfall of 700 mm [23]. Sampling was directed towards individuals with symptoms associated with genus *Macrophomina*; all samples were kept in plastic bags in a cooler until they were transferred to laboratory, to be processed.

The samples were cut into small 5 mm pieces, disinfected with 1% sodium hypochlorite for 3 min, and washed with sterile water. Finally, they were wrapped with sterile paper towels and placed in a laminar flow chamber at 20 °C for 15 min [24]. Subsequently, the samples were placed upright in Petri dishes with potato dextrose agar medium (PDA, Dioxon) modified with chloramphenicol (20 mg/mL^−1^) and incubated at 28 °C for 5 days. The identification of fungal colonies associated with the genus *Macrophomina* was carried out by the observation of reproductive structures under a microscope and employing taxonomic keys of Barnett and Hunter [25]. For the microscopic observation, thin layer PDA cultures (microculture technique) were used. The mycelial cultures were observed after eight days of incubation at 28 °C employing lactophenol. The microscopic morphology of the fungi was examined under an optic microscope (Carl Zeiss, Jena, Germany) [26].

### 2.2. DNA Extraction, PCR Amplification, and Sequencing

This procedure was performed with the 2% cetyl trimethylammonium bromide (CTAB) method according to Doyle and Doyle [27] with some modifications [28]. Genomic DNA was suspended in 100 µL of sterile HPLC water and quantified by spectrophotometry in a NanoDrop 2000c (Thermo Scientific, Waltham, MA, USA). To determine the DNA quality, absorbance values between 1.8 and 2.2 at A_280_/_260_ and A_230_/_260_ nm were considered acceptable. Finally, the DNA was diluted to 20 ng µL^−1^ and then stored at −20 °C for PCR amplification.

Molecular identification of “PUE 4.0” strain was carried out based on the analysis of internal transcribed spacer (ITS) region sequences using primer pairs ITS5 (5′-GGAAGTAAAAGTCGTAACAAGG-3′)/ITS4(5′-TCCTCCGCTTATTGATATGC-3′) [29]. The reaction mixture was prepared in a final volume of 15 µL with 1× Taq buffer DNA polymerase, 0.18 µM of each dNTP, 0.18 µL of each primer containing 10 pmol, 0.90 U of GoTaq DNA polymerase (Promega, Madison, WI, USA), and 40 ng µL^−1^ DNA. PCR was performed in a Peltier PTC-200 DNA thermal cycler (Bio-Rad, Santa Rosa, CA, USA). The amplicons were verified by electrophoresis in a 1.5% agarose gel (Seakem, Invitrogen, Carlsbad, CA, USA) and stained with 10,000× GelRed (Biotium, Fremont, CA, USA). All PCR products were cleaned with ExoSAP-IT (Affymetrix, Santa Clara, CA, USA), and both strands were individually sequenced using the BigDye Terminator v3.1 Cycle Sequencing Kit (Applied Biosystems, Carlsbad, CA, USA) in a 3130 Genetic Analyzer Sequencer (Applied Biosystems, Carlsbad, CA, USA) at Postgraduate College Facilities, Mexico, according to Juárez-Vázquez [30]. Sequences were assembled and edited using SeqMan (DNAStar, Madison, WI, USA) and compared to sequences established in GenBank™ using the Blast algorithm.

### 2.3. Pathogenicity Tests

Fifty peanut plants of the “Virginia Champs” variety provided by local farmers were used. Each two-month-old plant was individually planted in a 1 L plastic pot, containing a sterilized mixture of Peatmoss and Agrellite (1:1 *v*/*v*). The inoculation of “PUE 4.0” strain (*M. phaseolina*) was by the toothpick method three days after sowing [31]. Development of the plants was done under greenhouse conditions (70% relative humidity and 28 °C) until the appearance of symptoms of disease.

Toothpicks were previously sterilized and then placed in Petri dishes with the *M. phaseolina* colony until 100% colonization was reached. Using toothpicks with mycelium, small 2 mm wounds were made on the roots. Sterile toothpicks were used for control group plants, tests were carried out in duplicate. After three weeks, inoculated plants showed symptoms of wilting chlorosis on leaves and discoloration of the vascular ring, from brown to dark brown: characteristic symptoms of charcoal rot, while control plants remained healthy, fulfilling Koch’s postulates.

### 2.4. In Vitro Assessment of Antagonistic Capacity of Trichoderma spp.

For the evaluation of the antagonism test in vitro, the group of strains of *T. harzianum* (T-H3), *T. asperellum* (T-AS1), *T. hamatum* (T-A12), *T. koningiopsis* (T-K11), and an endemic strain of the *T. harzianum* (T-Ah) study region were used, whose sequences are found in the database of the National Center for Biological Information (NCBI) with access numbers MK780094, MK778890, MK791650, and MK791648, respectively. The dual confrontations were carried out with the strain “PUE 4.0” of *M. phaseolina* (MW585378) in a completely randomized experimental design with five treatments and three repetitions in duplicate.

For the evaluation of mycelial development, 5 mm diameter fragments of the *Trichoderma* strains as well as *M. phaseolina* were inoculated in Petri dishes with PDA (Potato and Dextrose Agar) and incubated in dark conditions at 28 °C for 10 days. The diameter of the mycelium was measured every 12 h with a digital vernier (CD-6 Mitutoyo) to estimate the growth speed (cm/d^−1^), which was calculated with the linear growth function [32] Equation (1).
y = mx + b(1)
where: y = is the distancem = slopex = is timeb = the constant factor.

Antagonism and percentage of inhibition were evaluated considering the mycelial growth radius of *Trichoderma* spp. and *M. phaseolina* (with their respective controls). PDA discs (5 mm in diameter) with mycelia of *Trichoderma* spp. and *M. phaseolina* were placed at the extremes of Petri plates containing PDA and incubated at 28 °C for 240 h. Then, mycelial growth was scored every 12 h until the first contact between the mycelia of each antagonist with *M. phaseolina* occurred [33].

The percentage of radial growth inhibition (PIRG) was calculated based on the formula of Equation (2).
PIRG% = (R1 − R2)/R1 × 100(2)
where: PIRG = Percent inhibition of radial growth.R1 = Radial growth (mm) of *M. phaseolina* without *Trichoderma* spp.R2 = Radial growth (mm) of *M. phaseolina* with *Trichoderma* spp.

Invasion of the antagonist or colonization on the surface of the *M. phaseolina* mycelium was taken as the index of antagonism with the scale proposed by Bell [34] (Table 1).

### 2.5. Field Experiment: Evaluation of Antagonism

A test was carried out under open field conditions in the community of Buenavista de Benito Juárez (18°27′39″ N; 98°37′11″ W), belonging to the municipality of Chietla in the state of Puebla-Mexico. Nine hundred and sixty (960) “Virginia Champs” variety peanut seedlings provided by community producers were used.

Community producers carried out land preparation three months before the establishment of the crop. First, a 50-cm-deep plow was made to reduce compaction and promote soil drainage, then two 30-cm-deep turns of earth were made to promote aeration ground. The transplant date was 7 July 2020, where peanut seedlings of 30 days of emergence were used, and they were sown at a depth of 8 cm. Finally, they were fertilized with 40 kg/ha of phosphorus (P) and 60 kg/ha of potassium (K) at 15 days after transplantation (dft). The population density was four plants per m^2^ spaced at 35 cm each and distributed in 26 rows with 60 cm between the rows of the crop, where two rows were considered to form an experimental block in a straight line.

The experimental design consists of 13 randomized complete blocks with 8 repetitions per treatment, occupying 100 plants per treatment, leaving four plants on the banks, which were not considered, giving a total of 800 plants of the 960 seedlings planted in the study community.

Inoculation of *M. phaseolina* (MW585378) was performed 15 days (dft) on the neck of each of the peanut plants (100 seedlings per treatment) with 1 mL of solution at a concentration of 1 × 10^8^ conidia. After 36 h, the plants were inoculated with strains of *T. harzianum* (T-H3), *T. asperellum* (T-AS1), *T. hamatum* (T-A12), *T. koningiopsis* (T-K11), and *T. harzianum* (T-Ah) at the same concentration as the pathogen (1 × 10^8^ conidia mL^−1^), for each treatment. For chemical treatment, Cercobin^®^ (Thiophanate methyl) was applied, following the manufacturer’s recommendations (500 g in 400 L^−1^ of water per ha). Finally, for the control treatment, only sterile water without the presence of fungal activity was applied.

The incidence of the disease expected by *M. phaseolina* was calculated in 100 plants per treatment at the end of four months that the cultivation lasted. For this, the infected portion was measured in relation to the total length of the roots [35] and it was classified on the scale proposed by Bokhari [36] where: 0 ≤ 25% severity, 1 = 26 to 50% severity, 2 = 51 to 75% severity, and 3 ≥ 76% severity. Additionally, the complementary variables of total fresh weight of each plant, dry weight of peanut pods per plant, number of pods per plant, and weight of 100 peanut grains per treatment were taken; in addition, the yield was calculated according to Zamurrad [37] at the end of experiment.

### 2.6. Statistical Analysis

Data were analyzed with ANOVA (two ways) in the statistical package SPSS Statistics version 17 for Windows. Growth speed and the development rate were response variables with three repetitions. Experiments were validated in duplicate in a completely randomized statistical design. The data were subjected to the Bartlett homogeneity test, and subsequently, a Tukey–Kramer comparison test of means was performed with a probability level of *p* ≤ 0.05.

Radial growth inhibition data (PIRG) were expressed in percentages and transformed with angular arccosine √x + 1. The mean values of the variables, which were the total fresh weight of the plant, dry weight of pods per plant, number of pods per plant, weight of 100 peanut grains, and yield were subjected to an analysis of variance with the same statistical program, using the test of Tukey–Kramer to determine the significant differences between the treatments (*p* < 0.05).

## 3. Results

### 3.1. Isolation, Characterization, and Identification of the Causative Agent of Charcoal Rot

Representative isolates from 50 different plants developed typical morphological characteristics of *M. phaseolina*. The fungal colonies were initially whitish grayish-dark brown in color on PDA medium (Figure 1a). After 6 days, semi-compressed mycelium was observed on the culture plate with microsclerotia embedded within the hyphae and absorbed into the agar. The aggregation of hyphae formed jet-black microsclerotia with a size of 74 × 110 μm (Figure 1b).

Amplification of 5.8S rDNA gene region showed a product of 601 bp, which presented 100% identity with *M. phaseolina* (ID: KF951698) in the Gen Bank nucleotide database of the National Center for Biotechnology Information (https://www.ncbi.nlm.nih.gov/, accessed on 15 February 2021). This sequence was deposited in the same database, with the accession number MW585378.

### 3.2. Pathogenicity Tests

Koch’s postulates confirmed that “PUE 4.0” strain showed typical symptoms of charcoal rot 20 days after inoculation. In addition, microsclerotia were observed in the vascular system (Xylem) which caused an upward wilt of stem, as shown in Figure 2b. No symptoms were observed in the control group.

Using PCR of repetitive sequence, it was possible to confirm the identity of the re-isolation (PUE 4.1) of the original strain. This sequence was deposited in the same database, with the accession number MW585379.

### 3.3. Percentage of Growth Inhibition In Vitro

Areas of interaction were observed between *T. harzianum* (T-H3), *T. asperellum* (T-AS1), *T. hamatum* (T-A12), *T. koningiopsis* (T-K11), and the native strain of *T. harzianum* (T-Ah) against *M. phaseolina* (MW585378), where parasitism greater than 50% was obtained at 240 h.

Development rate and the growth speed had significant differences (*p* ≤ 0.05), where *T. koningiopsis* (T-K11) obtained the highest value (Table 2) with 2.18 ± 0.035 mm/hour and 2.23 ± 0.013 cm/d^−1^, respectively. *M. phaseolina* showed the lowest growth speed (1.67 ± 0.054 cm/d^−1^).

The percentage of inhibition of radial growth (PIRG) presents significant differences (*p* = 0.0001). The double confrontation between the different *Trichoderma* species showed an inhibition greater than 50% from the tenth day (Table 2). However, the highest percentage of inhibition was obtained with *T. koningiopsis* (T-K11) obtaining 71.11%. Similarly, *T. harzianum* (T-H3) presented the second-best inhibition with 63.55%; both antagonistic strains presented a class I classification (Figure 3) according to the scale established by Bell et al. [34].

### 3.4. Evaluation of Antagonism in Field

Field results showed that the treatments with antifungal activity were effective in reducing the incidence of charcoal rot in the root, stem, and pods of “Virginia Champs” variety peanut plants under induced infection (Table 3).

The peanut plants that were inoculated with *M. phaseolina* at the time of transplantation presented root rots at 110 days; necrosis and elongated reddish-brown lesions were observed that covered 40% of the total surface of the main root (Figure 4a). In addition, a considerable loss of secondary roots, with abundant dark mycelium that formed black and rounded microsclerotia (Figure 4a) as well as necrosis in peanut pods, affecting their development, was observed (Figure 4b).

Inoculation with *M. phaseolina* caused a reduction in the height of the plant, presenting significant differences (*p* = 0.0056). It was possible to corroborate that *T*. *koningiopsis* (T-K11) was more effective in promoting the total growth of the plant, which reached a weight of 1417.60 g as well as reduced the incidence of charcoal rot by 76% (Table 3), compared to *T. harzianum* (T-H3), *T. asperellum* (T-AS1), *T. hamatum* (T-A12), and the strain native of *T. harzianum* (T-Ah).

Chemical treatment (Cercobin^®^) presented a 51% reduction of the incidence of charcoal rot without presenting significant differences with *T. harzianum* (T-H3), *T. asperellum* (T-AS1), *T. hamatum* (T-A12), and the strain native of *T. harzianum* (T-Ah) in comparison with plants inoculated only with *M. phaseolina* (Table 3). This can be explained in the following way: the *M. phaseolina* strain is native to the study region and may have acquired resistance towards Cercobin^®^, which is why the chemical treatment is no longer as effective to control charcoal rot.

The potential yield per hectare showed highly significant differences between the treatments (*p* = 0.001), where *T. koningiopsis* (T-K11) showed higher production (1.60 ± 0.01 t/ha^−1^), presenting 64.04 pods per m^2^. The chemical treatment (Cercobin^®^) was characterized by generating the second highest production, obtaining 60.88 pods per m^2^. The lowest production occurred in plants inoculated only with *M. phaseolina* (0.67 ± 0.01 t/ha^−1^), presenting 44 pods per m^2^. The average performance of the yields for the remaining strains of *Trichoderma* spp. oscillate between 1.28 ± 0.01 and 1.38 ± 0.01 ton ha^−1^ (Figure 5). If we consider the average yield per hectare in the region (1.30 t/ha^−1^), the results reported in the present investigation are superior to the treatments inoculated with biological agents, being able to generate a strategy in the future to control charcoal rot in the rural communities of Buenavista de Benito Juárez, which belongs to the municipality of Chietla in Puebla, Mexico.

## 4. Discussion

*M. phaseolina* is considered an important necrotrophic phytopathogen for various crops of agricultural interest, affecting at least 500 plant species in more than 100 families [12,38], where it has been reported to be a causal agent of charcoal rot, affecting the root and stem of peanut crops [9].

According to the morphological characteristics and the amplification of the 5.8S rDNA gene region [39], the identity of “PUE 4.0” strain was confirmed, corresponding to *M. phaseolina*, which coincides with the descriptions reported by Pandey [40] and Márquez [12].

The presence of microsclerotia in the Xylem caused the development of necrotic roots, chlorotic leaves, and premature death of the plants, symptoms that coincide with those reported in strawberry and sunflower crops [41]. This may be due to the presence of phytotoxic metabolites of *M. phaseolina* such as patulin, phaseolinon, and botryodiplodin, which play an important role in the early stages of charcoal rot [42,43]. As far as we know, this is the first report of this pathogen that causes charcoal rot in peanut (*Arachis hypogea* L.) crop in the variety “Virginia Champs” in Puebla, Mexico.

Species of the genus *Trichoderma*, known as green spore fungi, have been widely reported as biological control agents against diseases caused mainly by soil-borne pathogens [33].

In the present study, a parasitism greater than 50% was observed for all *Trichoderma* strains at 240 h against *M. phaseolina*, where there were areas of interaction in the dual culture assays. This may be due to the action capacity of the different *Trichoderma* species, as competition for space and nutrients [18] decreases the growth and development of the strain “PUE 4.0” in vitro.

Cubilla-Ríos [44] evaluated the antagonistic capacity of three *Trichoderma* species: *T*. *arundinaceum*, *T. brevicompactum*, and *T. harzianum* T34 on two isolates of *M. phaseolina* present in sesame and soybean, and found a greater antagonist activity exerted by *T. harzianum* T34. Likewise, Sreedevi [45] reported that *T. harzianum* reduced growth by 64.4% in *M. phaseolina*, isolated from the root rot of peanuts, results like those obtained in the present investigation with “T-H3” strain from *T. harzianum* (63.55%). According to the literature [46], the properties of *T. harzianum* can be attributed to the production of various volatile substances such as acetaldehyde; isocyanide derivatives, terpenes; derivatives of alpha-pyrone; piperazine, hydrazone derivatives as well as polyketides and alcohols [47]; and responsible for the degradation of the cell wall of fungi. This may be present in the inhibition process against *M. phaseolina* [48].

In Figure 2b,c, an interaction zone surrounding the mycelium of *M. phaseolina* could be observed. It could be rational to infer those volatile substances, enzymes that degrade the cell wall (Chitinase and Glucanase) and a large amount of antibiotics [49,50], exert a greater inhibition on the growth and development of “PUE 4.0” strain. In this sense, the strain of *T. koningiopsis* (T-K11) exerted a greater inhibition of radial growth (PICR) on *M. phaseolina* from the tenth day and a class I classification, according to the scale established by Bell [34]. Ruangwong [51] mentions that *T. koningiopsis* (PSU3-2) contains azetidine, 2-phenylethanol, and ethyl hexadecanoate; compounds that may be associated with antibiosis and suppression of growth on the mycelial of *M. phaseolina* in the present investigation.

The biological control of *M. phaseolina* under field conditions through different isolates of *Trichoderma* reduced the charcoal rot for the peanut crop. In addition, it was possible to observe reddish brown lesions in the total surface of the root in the group of plants inoculated only with “PUE 4.0” strain, characteristic symptoms of *M. phaseolina* mentioned by Ghosh [9] for the peanut crop, and Etebarian [52] for the melon crop.

Of the different microbial antagonists used in the present investigation, *T. koningiopsis* (T-K11) and *T. harzianum* (native) showed more efficient results in reducing charcoal rot and the incidence of the disease. The same finding was found by Hussain [22] and Khan [53], who observed that *T. harzianum* is effective in controlling *M. phaseolina* for mung beans. Likewise, Dubey [54] observed that *T. harzianum* increased the frequency of healthy plants, results similar to those obtained in the present investigation by *T. koningiopsis* (T-K11), which managed to reduce the incidence of charcoal rot by 76% in the field.

If we consider the average of China as the largest peanut producer in the world (1.45 t/ha^−1^), the average yield obtained in the present investigation with *T. koningiopsis* (T-K11) showed higher production (1.60 ± 0.01 t/ha^−1^) [3]. This can be explained by the fact that the genus *Trichoderma* can inhibit the growth of pathogenic fungi and exert positive effects on the absorption of nutrients, plant growth, and yield, in addition to protecting them against biotic and abiotic stress. [55,56,57]. In a study presented by Osman [58], it was observed that *T. harzianum* alone and combined with yeast improved the yield and quality of peanuts under field conditions. On the other hand, Dania [59], found the highest number of peanut pods per plant (15.67) in a combination of *T. hamatum* and cattle manure, results similar to those obtained in the present investigation by *T. koningiopsis* (16.01) and *T. harzianum* (15.35).

## 5. Conclusions

It was possible to identify *M. phaseolina* (MW585378 and MW585379) associated with charcoal rot from the crop of peanuts “Virginia Champs” variety, located in the rural communities of Buenavista de Benito Juárez, belonging to the municipality of Chietla in Puebla, Mexico.

The strains *T. koningiopsis* (T-K11) and *T. harzianum* (TH-3) displayed class I of antagonism on the Bell scale. In addition, *T. koningiopsis* (T-K11) displayed the highest rate of development, speed of growth, and percentage of inhibition of PIGR radial growth on *M. phaseolina* (71.11%) under in vitro conditions.

In field conditions, *T. koningiopsis* (T-K11) was more effective in promoting the total growth of the plant reaching a weight of 1417.60 g as well as in reducing the incidence of the disease by 76%.

In the locality of Buenavista de Benito Juárez, belonging to the municipality of Chietla in Puebla, Mexico, *T. koningiopsis* (T-K11) showed higher production (1.60 ± 0.01 t/ha^−^^1^) as well as the chemical treatment (Cercobin^®^), which obtained the second highest production, obtaining 64 pods per m^2^. Lowest production occurred in plants inoculated only with *M. phaseolina* (0.67 ± 0.01 t/ha) where elongated reddish-brown lesions were observed that covered 40% of the total surface of the main root.

## Figures and Tables

**Figure 1 plants-10-02630-f001:**
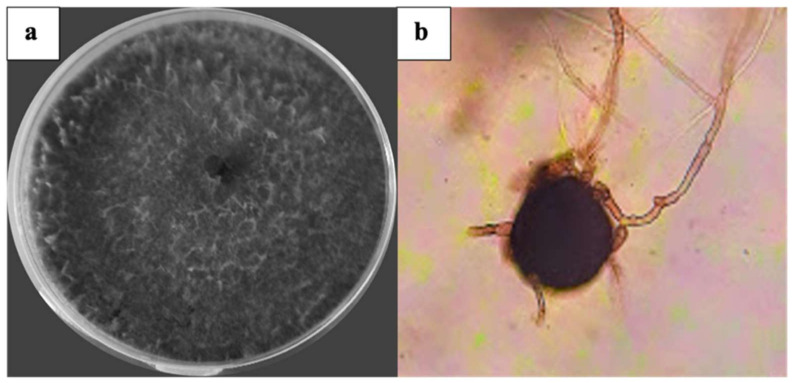
Cultural and morphological characteristics of *M. phaseolina.* (**a**) Colony on the agar plate; (**b**) microsclerotia (100×).

**Figure 2 plants-10-02630-f002:**
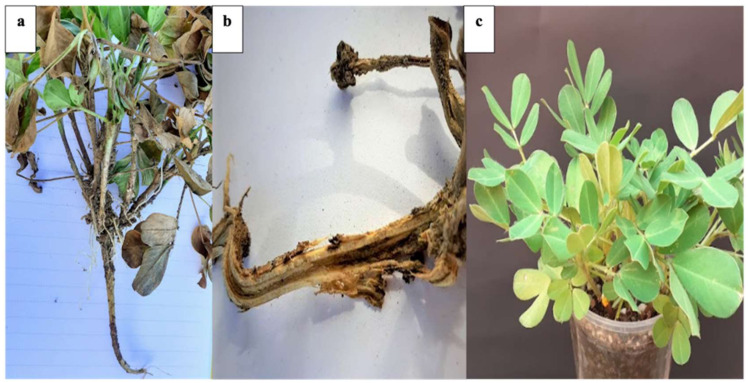
Pathogenicity tests with “PUE 4.0” strain: (**a**) plant of *Arachis hypogea* with charcoal rot and death of foliage at 20 days after inoculation; (**b**) cross-section of peanut root showing rot and the presence of microsclerotia; (**c**) control group without symptoms.

**Figure 3 plants-10-02630-f003:**
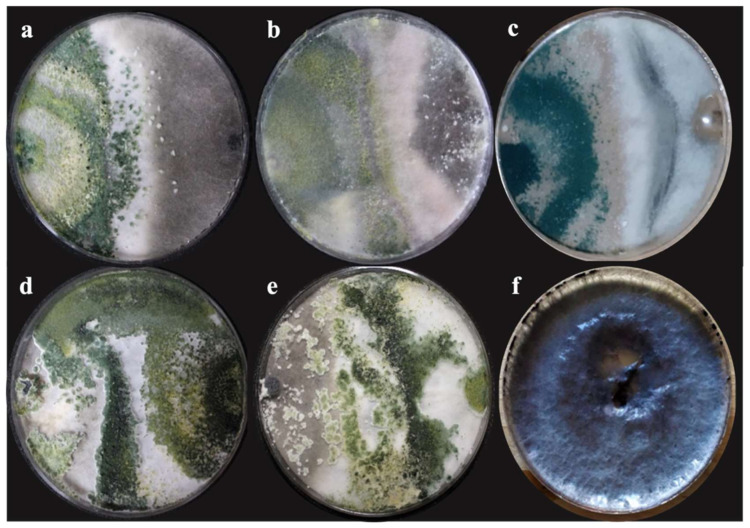
Antagonism of *T. asperellum* (**a**), *T. harzianum* native (**b**), *T. hamatum* (**c**), *T. koningiopsis* (**d**), and *T. harzianum* (**e**) with *M. phaseolina* on the scale by Bell et al. [34] after 132 h in dishes with PDA medium, incubated at 28 °C for 10 days. (**d**,**e**) Class I antagonism; (**a**–**c**) class II antagonism; (**f**) control group of *M. phaseolina*.

**Figure 4 plants-10-02630-f004:**
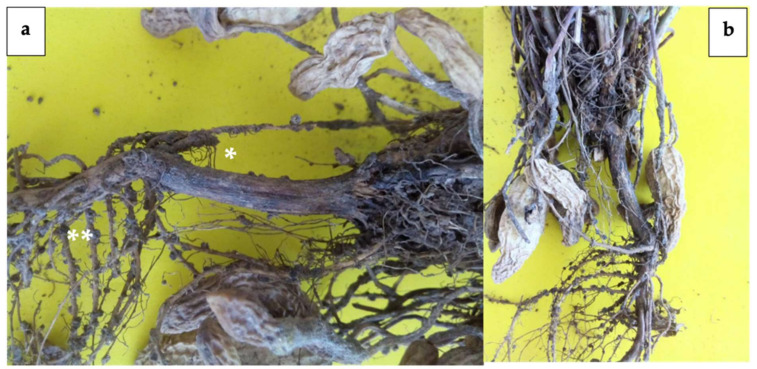
Charcoal rot in peanut plants that were inoculated with *M. phaseolina*. (**a**) reddish brown necrosis * present on the surface of the main root with abundant dark mycelium that formed black and rounded microsclerotia on the lateral roots **; (**b**) necrosis in peanut pods.

**Figure 5 plants-10-02630-f005:**
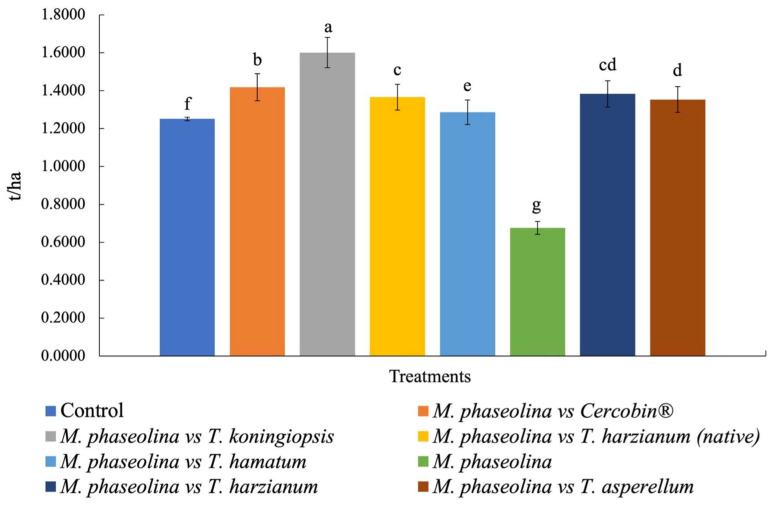
Potential yield (t/ha^−1^) of peanuts for each treatment in Buenavista de Benito Juárez, which belongs to the municipality of Chietla in Puebla, Mexico. Media followed by the same letter do not present significant statutory differences (*p* ≤ 0.05) according to Tukey’s test.

**Table 1 plants-10-02630-t001:** *Trichoderma* strain antagonism evaluated in vitro using Bell’s scale [34], considering the invasion of the surface.

Class	Class Features
I	*Trichoderma* completely overgrew the *M. phaseolina* and covered the entire medium surface.
II	*Trichoderma* overgrew at least two-thirds of the medium surface.
III	*Trichoderma* and the *M. phaseolina* each colonized approximately one-half of the medium surface and neither organism appeared to dominate the other.
IV	*M. phaseolina* colonized at least two-thirds of the medium surface and appeared to withstand encroachment by *Trichoderma*.
V	*M. phaseolina* completely overgrew the *Trichoderma* and occupied the entire medium surface.

**Table 2 plants-10-02630-t002:** Rate of development, growth speed, and percentage of inhibition of radial growth and antagonism classification on Bell Scale [34].

Name	Development Rate (mm/hour) *	Growth Rate (cm/d^−1^) *	PICR *	Class Antagonism
*T. harzianum* (T-H3)	2.17 ± 0.095 ^a^	1.32 ± 0.04 ^a^		
*T. asperellum* (T-AS1)	1.86 ± 0.033 ^b^	2.16 ± 0.017 ^a^		
*T. hamatum* (T-A12)	1.60 ± 0.05 ^c^	2.14 ± 0.01 ^a^		
*T. koningiopsis* (T-K11)	2.18 ± 0.035 ^a^	2.23 ± 0.013 ^a^		
*T. harzianum* (Th-Ah)	1.59 ± 0.04 ^c^	2.06 ± 0.068 ^a^		
*M. phaseolina* (PUE 4.0)	1.44 ± 0.04 ^c^	1.67 ± 0.054 ^b^		
*M. phaseolina* (PUE 4.0) vs. *T. harzianum* (T-H3)			63.55 ± 0.88 ^ab^	I
*M. phaseolina* (PUE 4.0) vs. *T. asperellum* (T-AS1)			53.33 ± 0.76 ^bc^	II
*M. phaseolina* (PUE 4.0) vs. *T. hamatum* (T-A12)			51.55 ± 2.47 ^c^	II
*M. phaseolina* (PUE 4.0) vs. *T. koningiopsis* (T-K11)			71.11 ± 0.44 ^a^	I
*M. phaseolina* (PUE 4.0) vs. *T. harzianum* (Th-Ah)			59.11 ± 4.23 ^bc^	II

* Media followed by the same letter do not present significant statutory differences (*p* ≤ 0.05) according to Tukey’s test. Means followed by the same letter (a, b and c) are not significantly different for *p* ≤ 0.05 according to Tukey test.

**Table 3 plants-10-02630-t003:** Antagonistic activity on the incidence of disease, the weight per plant, weight of 100 grains the peanuts, weight, and number of pods.

Treatments	Incidence of Disease	Total Fresh Weight per Plant (g) *	Dry Weight of Pods per Plant (g) *	Number of Pods per Plant *	Weight of 100 Grains (g) *
		M ± SE	M ± SE	M ± SE	M ± SE
*M. phaseolina*	3	673.20 ± 52.04 ^e^	54.80 ± 10.74 ^b^	11.00 ± 0.28 ^e^	58 ± 0.09 ^f^
*M. phaseolina* vs. *T. hamatum*	2	909.80 ± 62.6 ^c^	112.60 ± 14.05 ^a^	14.60 ± 0.04 ^c^	61 ± 0.02 ^e^
*M. phaseolina* vs. *T. asperellum*	2	970.60 ± 132.71 ^bc^	115.60 ± 7.83 ^a^	14.88 ± 0.61 ^bc^	62 ± 0.04 ^d^
*M. phaseolina* vs. *T. koningiopsis*	1	1417.60 ± 101.61 ^a^	124.20 ± 8.60 ^a^	16.01 ± 0.71 ^a^	64.8 ± 0.01 ^a^
*M. phaseolina* vs. *T. harzianum*	2	1018.60 ± 55.52 ^b^	116.20 ± 9.15 ^a^	15.35 ± 0.6 ^b^	63 ± 0.04 ^c^
*M. phaseolina* vs. *T. harzianum* (native)	2	1007.80 ± 74.28 ^bc^	123.00 ± 5.27 ^a^	15.13 ± 0.81 ^bc^	62.4 ± 0.04 ^cd^
*M. phaseolina* vs. Cercobin^®^	2	1077.00 ± 112.81 ^b^	120.60 ± 18.60 ^a^	15.22 ± 0.86 ^b^	64 ± 0.03 ^b^
Control	0	712.20 ± 105.86 ^d^	88.00 ± 7.62 ^b^	13.89 ± 0.52 ^d^	60.9 ± 0.04 ^e^

* Media followed by the same letter do not present significant statutory differences (*p* < 0.05) according to Tukey’s test. (*n* = 100 plants per treatment), M = mean, SE = Standard Error. Means followed by the same letter (a, b, c, d, e and f) are not significantly different for *p* ≤ 0.05 according to Tukey test.

## Data Availability

Informed consent was obtained from all subjects involved in the study.

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
