# Peer review of "Biological Control of Charcoal Rot in Peanut Crop through Strains of Trichoderma spp., in Puebla, Mexico"

_plants, 2021, doi:10.3390/plants10122630_

Round 1
Reviewer 1 Report
The object of the paper is of potential interest to scientists involved in biological control.
Unfortunately, the manuscript is quite unreadable because of poor English language ( for example see lines 72-74, line 93, line 148, lines 151-152, lines 190-191, etc).
The section Mat Meth is quite confusing ( see incidence definition line202-; lines 213-215; line 227).
Table 3 lacks basic information: numbers are mean values with SD or SE? Mean values of how many samples? Dry or fresh of total weight?
Figure 4 should be radically improved in order to be readable.
The discussion section should be drastically reduced and limited to the discussion of results. In the present form, it resembles a state-of-the-art review.
Author Response
Thanks for the comments, I also comment the following:
The manuscript fits very well with the scope of the journal's special edition (Plant Protection and Biotic Interactions), which highlights the biotic interactions of native strains of Trichoderma spp., Against an emerging peanut disease caused by Macrophomina paseolina in Mexico.
Likewise, I inform you that each of your comments were met to substantially improve the work presented. In the attached file, the changes marked in yellow color are shown.
Reviewer Comments
- Unfortunately, the manuscript is quite unreadable due to bad English language (for example, see lines 72-74, line 93, line 148, lines 151-152, lines 190-191, etc.).
Answer
The dramatic quality of the manuscript has improved markedly, and the suggested lines marked in yellow in the text are more clearly detailed.
It is worth mentioning that the work was reviewed by a native speaker of the English language (USA) to comply with this observation. If you think that this answer is not enough, I can tell you that we can opt for the style correction offered by the magazine; This is to comply with the language requirement, and not because of that, the language is an obstacle that prevents the publication of these results.
We believe that this document has technical-scientific relevance in Plant Protection in the cultivation of peanuts in the state of Puebla, Mexico.
Reviewer Comments
- The Mat Meth section is quite confusing (see incidence definition line 202; lines 213-215; line 227).
Answer
The changes were made. Suggested lines marked in yellow in the text are more clearly detailed.
Reviewer Comments
- Table 3 lacks basic information: are the numbers mean values ​​with SD or SE? Mean values ​​of how many samples? Full weight dry or fresh?
Answer
The requested information is included in Table 3.
Reviewer Comments
- Figure 4 needs to be radically improved to be readable.
Answer
The quality of figure 4 has been improved to make it legible
Reviewer Comments
- The discussion section should be drastically shortened and limited to discussion of the results. In its current form, it resembles a cutting-edge review.
Answer
The wording of the discussion section is improved, and the expected results are adequately discussed, thus supporting the objective of this research.

Reviewer 2 Report
This manuscript was difficult to read and follow due to poor English. The subject is interesting, the experiment is complete and well set and I am very sorry to reject the paper. But with such proper experimental design and interesting result, I expect from authors to present their work seriously and scientifically and professionally. For that reason, I put in detailed what has to be done to improve the manuscript. I encourage the authors to accept this suggestion and to give the manuscript to English native speaker with experience in scientific language to correct it and to resubmit again.
The abstract is awkwardly written, very confusing and in poor English. Not all important information are listed, especially regarding the goals of the investigation. For example, objectives regarding the field experiment are not mentioned at all, but later in the abstract there is an information about the productivity.
I suggest to completely overwrite the Abstract.
Keywords should be more specific with regard to the ms subject
Line 40: 17 tons in China? If it is so, the China is not the leader in production
Line 43: Crop of peanuts presents various diseases..?? What authors wanted to say with this? Maybe you mean that Peanut crop could be affected by various diseases caused by..
Line 46: Please, put the full name of bacteria when you mention it for the first time
Line 57: what do you mean by …shedding of the tissue critical of the lower stem…? Please describe with more clarity.
Line 62-65: Omit the sentence and change the next one to: Faced with the high costs of fungicides and their potentially harmful effects on people and environment, biological control is considered…
Line 72: stand out put at the end of the sentence
Line 74: …it still remains a challenge
Line 93: Later, samples were
Line 97: protocol of Morales?
Line 98: omit the word obtained
Lines 98-104: Please present it more simply
Line 101: compare to what?? Please rephrase this part of a sentence. This whole sentence is very complicated written and contains to much information. It should be divided into at least two.
Line 117: prepared
Line 133: Fifty two..what does it mean of 60 days? Is the official name Virginia Champs Variety or just Virginia Champs? How many plants did you inoculate?
Line 134: were inoculated using the toothpick method in a individual plastic pot…
Section 2.3. is not performed “in vitro” but “in vivo” since the plants were grown in the greenhouse. The only thing is that you infected the toothpicks in vitro, but whole experiment is performed in vivo. Please change that in the text.
In the section 2.4. the English is very bad and it is hard to understand the procedure. Please overwrite this section. I propose to name this section: In vitro assessment of antagonistic capacity of Trichoderma spp.
Line 159: what is m and please elaborate the equation clearly
Line 183: Instead of 960, write the number,
transpant date?, what was fertilized?
Lines 200 and 201: ..sterile water was applied..
Section 2.5. it is not clear how you set up your experiment: one treatment only with M. phaseolina, 5 with different strains of Trichoderma, one chemical treatment, one healthy control free of M. phaseolina and Trichoderma, 100 plants per treatment, 8 repetitions and how do we come to the 960 plants?
2.6. Please correct English
Line220: variance analysis, as well as the data on fresh weight..
Line276: instead against put and, after 240 h
In the Table 3 change the descriptive title to include other parameters form table
Line 315: lateral roots
Line 335: Explain the 51% reduction
In the Result section, correct English style.
How do you explain that control plants achieved less yield then infected plants?
The discussion should be overwritten. As it is, does not discuss the gained results from this experiment and it is the main objective. The authors should confront their results to the results of other authors on the subject, and not mainly to discuss other scientist's results. So, in the main focus of discussion should be their own results from this investigation. The text from the discussion as it is now is good and the authors can use it to create new discussion with focus on their own results.
Author Response
Thanks for the comments, I also comment the following:
The manuscript fits very well with the scope of the journal's special edition (Plant Protection and Biotic Interactions), which highlights the biotic interactions of native strains of Trichoderma spp. against an emerging peanut disease caused by Macrophomina paseolina in Mexico.
Likewise, I inform you that each of your comments were met to substantially improve the work presented, presented in green.
It is worth mentioning that the work was reviewed by a native speaker of the English language (USA) to comply with this observation. If you think that this answer is not enough, I can tell you that we can opt for the style correction offered by the magazine; This is to comply with the language requirement, and not because of that, the language be an obstacle that prevents the publication of these results.
We believe that this document has technical-scientific relevance in Plant Protection in the cultivation of peanuts in the state of Puebla, Mexico.
Considerations:
- The abstract is awkwardly written, very confusing and in poor English. Not all important information are listed, especially regarding the goals of the investigation. For example, objectives regarding the field experiment are not mentioned at all, but later in the abstract there is an information about the productivity.
Answer
The abstract is more clearly detailed in green, in addition, the objectives of the research work are incorporated into the text.
- Keywords should be more specific with regard to the ms subject.
Answer
Keywords are modified at the reviewer's suggestion.
- Line 40: 17 tons in China? If it is so, the China is not the leader in production.
Answer
It is verified that China is the largest peanut producer and the information is incorporated in the document marked in green.
- Line 43: Crop of peanuts presents various diseases..?? What authors wanted to say with this? Maybe you mean that Peanut crop could be affected by various diseases caused by....
Answer
The text is modified as suggested by the reviewer, marked in green.
- Line 46: Please, put the full name of bacteria when you mention it for the first time.
Answer
The text is modified as suggested by the reviewer, marked in green.
- Line 57: what do you mean by …shedding of the tissue critical of the lower stem…? Please describe with more clarity.
Answer
The text is modified as suggested by the reviewer, marked in green.
- Line 62-65: Omit the sentence and change the next one to: Faced with the high costs of fungicides and their potentially harmful effects on people and environment, biological control is considered…
Answer
The text is modified as suggested by the reviewer, marked in green.
- Line 72: stand out put at the end of the sentence.
Answer
Threw out
- Line 74: …it still remains a challenge.
Answer
The text is modified.
The quality of the manuscript has been remarkably improved as suggested by the reviewer, marked in green.
- Line 93: Later, samples were.
Answer
The text is modified.
- Line 97: protocol of Morales?
Answer
The text are more clearly detailed.
- Line 98: omit the word obtained.
Answer
Threw out
- Lines 98-104: Please present it more simply.
Answer
The text are more clearly detailed.
- Line 101: compare to what?? Please rephrase this part of a sentence. This whole sentence is very complicated written and contains to much information. It should be divided into at least two.
Answer
The text is modified and detailed with greater clarity.
- Line 117: prepared.
Answer
Threw out
- Line 133: Fifty two..what does it mean of 60 days? Is the official name Virginia Champs Variety or just Virginia Champs? How many plants did you inoculate?
Answer
The text is modified and detailed with greater clarity.
The variety used is Virginia Champs.
60 days means that the plants used are approximately two months old.
- Line 134: were inoculated using the toothpick method in a individual plastic pot…
Answer
The text is modified and detailed with greater clarity.
- Section 2.3. is not performed “in vitro” but “in vivo” since the plants were grown in the greenhouse. The only thing is that you infected the toothpicks in vitro, but whole experiment is performed in vivo. Please change that in the text.
Answer
The text is modified.
- In the section 2.4. the English is very bad and it is hard to understand the procedure. Please overwrite this section. I propose to name this section: In vitro assessment of antagonistic capacity of Trichoderma spp.
Answer
The text is modified and detailed with greater clarity.
- Line 159: what is m and please elaborate the equation clearly
Answer
The text is modified and detailed with greater clarity.
- Line 183: Instead of 960, write the number.
Answer
The text is modified and detailed with greater clarity.
- transpant date?, what was fertilized?
Answer
The text is modified and detailed with greater clarity.
- Lines 200 and 201: ..sterile water was applied...
Answer
In the control treatment, only sterilized water was applied without the presence of fungal activity.
- Section 2.5. it is not clear how you set up your experiment: one treatment only with M. phaseolina, 5 with different strains of Trichoderma, one chemical treatment, one healthy control free of M. phaseolina and Trichoderma, 100 plants per treatment, 8 repetitions and how do we come to the 960 plants?
Answer
The text is modified and detailed with greater clarity.
The experimental design consists of 13 randomized complete blocks with 8 repetitions per treatment, occupying 100 plants per treatment, leaving four plants on the banks, which were not considered, giving a total of 800 plants of the 960 seedlings planted in the study community.
- 2.6. Please correct English
Answer
The text is modified and detailed with greater clarity.
- Line 220: variance analysis, as well as the data on fresh weight..
Answer
The text is modified and detailed with greater clarity.
- Line276: instead against put and, after 240 h.
Answer
The text is modified and detailed with greater clarity.
- In the Table 3 change the descriptive title to include other parameters form table.
Answer
Threw out
- Line 315: lateral roots.
Answer
The text is modified and detailed with greater clarity.
- Line 335: Explain the 51% reduction.
Answer
Threw out
This can be explained in the following way: The M. phaseolina strain is native to the study region and may have acquired resistance towards Cercobin®, which is why the chemical treatment is no longer as effective to control charcoal rot.
- In the Result section, correct English style.
Answer
The text is modified and detailed with greater clarity.
- How do you explain that control plants achieved less yield then infected plants?
Answer
This can be explained by the fact that the genus Trichoderma can inhibit the growth of pathogenic fungi and exert positive effects on the absorption of nutrients, plant growth and yield, in addition to protecting them against biotic and abiotic stress.
- The discussion should be overwritten. As it is, does not discuss the gained results from this experiment and it is the main objective. The authors should confront their results to the results of other authors on the subject, and not mainly to discuss other scientist's results. So, in the main focus of discussion should be their own results from this investigation. The text from the discussion as it is now is good and the authors can use it to create new discussion with focus on their own results.
Answer
The text is modified and detailed with greater clarity.
in addition, the results obtained are better discussed as suggested by the reviewer.

Round 2
Reviewer 1 Report
Overall improvements are negligible.
I really doubt that the manuscript has been revised by a native English-speaking person.
Reviewer 2 Report
In this form, the manuscript could be published. It is much improved after the revision.